# Uncertainty-Aware Brain Tumor Segmentation Using Attention Residual U-Net with Guided Decoder and Monte Carlo Dropout

## Abstract

Precise brain tumor segmentation from MRI scans is essential for successful diagnosis, treatment planning, and follow-up. Through this research, we developed a new model, ARU-GD+MCD by incorporating Monte Carlo Dropout layers into the Attention Res-UNet with Guided Decoder (ARU-GD), a state-of-the-art architecture for brain tumor segmentation. These dropout layers enable the model to estimate uncertainty by capturing variability during prediction, thereby generating uncertainty maps that indicate regions of low confidence in the segmentation results. This provides doctors with informative insights into the reliability of the model's outputs. Tested on the BraTS 2019 dataset with four MRI modalities (FLAIR, T1, T1CE, T2), the improved ARU-GD attained Dice scores of 0.886, 0.899, and 0.856, and IoU scores of 0.793, 0.818, and 0.748 for whole tumor, tumor core, and enhancing tumor regions, respectively. Our method compares favorably to baseline models such as UNet and Res-UNet not only in terms of segmentation accuracy but also through the addition of an important interpretability layer. These advances enable more confident and better-informed clinical decision-making, ultimately leading to improved patient outcomes.

## 1 Introduction

Brain tumor segmentation from MRI scans plays a critical role in diagnosis, treatment planning, and follow-up care. In clinical settings, even with the aid of automated segmentation models, doctors manually analyze the complete predicted tumor regions to finalize the tumor boundaries. This manual inspection is time-consuming and can delay treatment decisions. Various deep networks have been applied for brain tumor segmentation Havaei et al. (2017).

While recent work has focused heavily on improving segmentation accuracy, there still exists a gap because doctors must manually review the entire predicted segmentation. This is necessary since they do not know which parts of the prediction are correct and which are uncertain. Without information about the model's confidence in different regions, doctors cannot fully trust the automated results, leading to increased time spent on verification and potential risks to patient care.

To meet this challenge, we have developed a solution that not only provides precise tumor segmentation but also generates uncertainty maps indicating the model's confidence across various regions of the output. These maps give doctors a quick and intuitive sense of which areas of the prediction are reliable and which may require closer inspection. By highlighting areas of low confidence, uncertainty maps minimize the time doctors devote to manual verification and enable them to focus on the most critical details of the scan, leading to enhanced diagnostic efficiency and improved patient care Jungo & Reyes (2020).

Our work improves upon the current Attention Res-UNet with Guided Decoder (ARU-GD) model by incorporating Monte Carlo dropout layers. This addition enables the model to learn predictive uncertainty and produce useful uncertainty estimates during inference. Our approach bridges the gap between high accuracy and real-world clinical usability, enabling more informed, faster, and safer medical decisions.

## 2 OVERVIEW

### 2.1 ARU-GD ARCHITECTURE

The Attention Res-UNet with Guided Decoder (ARU-GD) is an effective deep learning model for medical image segmentation. It extends the U-Net architecture for better segmentation accuracy Ronneberger et al. (2015)Oktay et al. (2018)Long et al. (2015) by adding residual connections, attention gates, and a guided decoder. Unlike automated and self-configuring segmentation frameworks such as nnU-Net Isensee et al. (2021), ARU-GD incorporates explicitly designed architectural enhancements tailored for medical imaging tasks. The residual blocks assist in training deeper networks by preventing vanishing gradient problems, while the attention gates allow the model to highlight important spatial features Oktay et al. (2018) by suppressing unimportant background noise Woo et al. (2018). Spatial attention also improves segmentation around tumour edges Guo et al. (2021). The guided decoder improves feature learning by controlling intermediate layers, enhancing the quality of the final segmentation. ARU-GD demonstrates excellent performance in medical segmentation tasks, especially in brain tumor segmentation Maji et al. (2020).

### 2.2 BAYESIAN UNCERTAINTY AND MONTE CARLO DROPOUT

Traditional deep learning models often provide a single deterministic output without indicating how confident the model is in its prediction. This can be problematic in critical applications like healthcare. To address this, Bayesian deep learning introduces the concept of modeling uncertainty. Bayesian neural networks enable modeling uncertainty through probabilistic weights Konathala (2019), often implemented via variational inference techniques Graves (2011). One practical approach to approximate Bayesian inference in neural networks is using Monte Carlo dropout Gal & Ghahramani (2015), though other approaches like auto-encoding variational inference have also been proposed Kingma & Welling (2013).

Monte Carlo dropout serves as a practical approximation to Bayesian inference Kendall & Gal (2015). Other techniques such as variational dropout with local reparameterization have also been introduced for more efficient uncertainty modeling Molchanov et al. (2017)Kingma et al. (2015) in Monte Carlo, dropout layers (normally used during training to prevent overfitting) are also kept active during inference. By passing the same input through the model multiple times with dropout enabled, a distribution of predictions is obtained. From this distribution, predictive uncertainty can be quantified, typically using metrics like predictive entropy. This enables capturing epistemic uncertainty Blundell et al. (2015), uncertainty arising due to limited training data or model knowledge, making the approach especially suitable for high-stakes domains like medical imaging Gal & Ghahramani (2016). Moreover, distinguishing between uncertainties that stem from data variability (aleatoric) and those due to model limitations (epistemic) is crucial for robust decision-making in such sensitive contexts Kendall & Gal (2017). This uncertainty information helps highlight areas in the model output where it is less confident-crucial for medical applications where interpretability and trust are essential.

## 3 METHODOLOGY

### 3.1 DATASET SELECTION

This work utilized the BraTS 2019 dataset, an official dataset used for brain tumor segmentation. There are 259 high-grade glioma (HGG) patient cases with four types of MRIs (FLAIR, T1, T1CE, T2) and ground truth segmentation masks per case. Each MRI and mask are a 3D volume of size $155 \times 240 \times 240$ pixels, with four-class labels: background (0), tumor core (1), edema (2), and enhancing tumor (4). The data was split into 188 train patients (4700 slices), 31 validation patients (775 slices), and 40 test patients (1000 slices) to make an unbiased evaluation. The BraTS challenge provides a benchmark for tumor segmentation and has been instrumental in advancing research in this field Bakas et al. (2018)Menze et al. (2015).

## 3.2 PREPROCESSING

The MRI data was prepared by selecting 25 slices from the middle of each patient's 3D volume (indices 50 to 98) to focus on key tumor areas and reduce processing needs. The four MRI types were combined into a 4-channel input of shape 25×240×240×4. Each channel was normalized using Z-score normalization, calculated as:

$$x_{norm} = \frac{x - \mu}{\sigma + \epsilon} \tag{1}$$

where $x$ is the pixel intensity, $\mu$ is the mean, $\sigma$ is the standard deviation, and $\epsilon = 10^{-8}$ avoids division by zero. Ground truth masks were converted to a 4-channel format (25×240×240×4) matching the four classes. Preprocessed data was saved in HDF5 files to avoid repeating the preprocessing for each run, with separate files for training, validation, and test sets.

## 3.3 MODEL ARCHITECTURE

The Attention Res-UNet with Guided Decoder (ARU-GD) is a deep learning model designed for brain tumor segmentation, utilizing a U-Net-based architecture enhanced with residual connections, attention mechanisms, and guided decoding to improve feature extraction and segmentation accuracy. The model processes multi-modal MRI images of size 240×240×4, corresponding to four modalities (FLAIR, T1, T1ce, and T2), and produces four segmentation maps of size 240×240×4, representing four tumor classes (background, necrotic core, edema, and enhancing tumor). The architecture comprises an encoder, a bridge, a decoder with attention gates, dropout layers for uncertainty estimation, and multi-stage outputs for deep supervision.

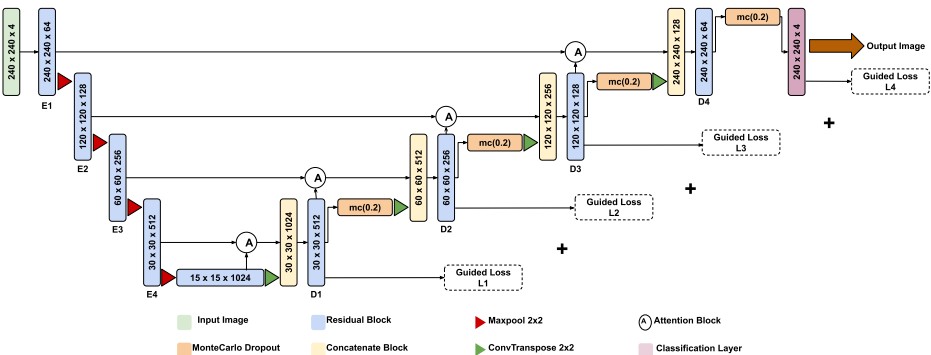

Figure 1: Aru-GD+MCD model architecture.

### 3.3.1 ENCODER

The encoder extracts hierarchical features from the input through a series of downsampling residual blocks, progressively reducing spatial dimensions while increasing the depth of feature representations. As shown in Figure 1 each encoder block applies two 3×3 convolutional layers with batch normalization and LeakyReLU activation ($\alpha = 0.2$), followed by a residual connection using a 1×1 convolution to preserve information flow. Max-pooling (2×2, stride 2) is applied between blocks to downsample the spatial dimensions by a factor of 2, enabling the capture of multi-scale contextual information. The encoder consists of four blocks with the following output dimensions:

- E1: 240×240×64
- E2: 120×120×128
- E3: 60×60×256
- E4: 30×30×512

The encoder's output at each stage is passed to the corresponding decoder block via skip connections, facilitating the integration of low-level and high-level features during upsampling.

### 3.3.2 BRIDGE

The bridge serves as the bottleneck, connecting the encoder and decoder paths, and processes the output of the final encoder block to produce the deepest feature representation, capturing global contextual information. It applies max-pooling ($2\times2$, stride 2) to downsample the spatial dimensions, followed by a residual block consisting of two $3\times3$ convolutional layers with batch normalization, LeakyReLU activation ($\alpha = 0.2$), and a $1\times1$ convolution for the residual connection to ensure stable gradient flow. The bridge output dimension is:

- Bridge: 15x15x1024

This deep representation is then passed to the decoder for upsampling and refinement.

### 3.3.3 DECODER

The decoder reconstructs the segmentation map by upsampling the bridge output and integrating features from the encoder via attention gates. Each decoder block upsamples the feature maps using $2\times2$ transposed convolutions (stride 2), doubling the spatial dimensions while reducing the number of filters. Dropout layers with a rate of 0.2 are applied after the residual block in each decoder stage (D1 to D4). Attention gates focus on relevant encoder features by reducing them to intermediate filter sizes (256 for D1, 128 for D2, 64 for D3, 32 for D4) using $2\times2$ (stride 2) and $1\times1$ convolutions, followed by additive fusion, LeakyReLU ($\alpha = 0.2$), a $1\times1$ convolution, sigmoid activation, upsampling, and element-wise multiplication with encoder features. The upsampled and attention-weighted features are concatenated and processed through a residual block (two $3\times3$ convolutions with batch normalization and LeakyReLU), followed by dropout to enable uncertainty estimation. The decoder consists of four blocks with the following output dimensions:

- D1: $30\times30\times512$ (dropout applied)
- D2: $60\times60\times256$ (dropout applied)
- D3: $120\times120\times128$ (dropout applied)
- D4: $240\times240\times64$ (dropout applied)

### 3.3.4 GUIDED DECODER AND MULTI-STAGE OUTPUTS.

The guided decoder generates intermediate segmentation maps ($out_1$, $out_2$, and $out_3$) from decoder stages D1, D2, and D3, respectively. This provides deep supervision, which improves gradient flow during training. These intermediate outputs are upsampled to the input resolution of $240\times240$ and processed through a $1\times1$ convolution with a softmax activation to produce segmentation maps of size $240\times240\times4$. The upsampling steps for each map are as follows:

- $out_1$ (from D1): Upsampled through three steps
  ($30\times30 \rightarrow 60\times60 \rightarrow 120\times120 \rightarrow 240\times240$)
- $out_2$ (from D2): Upsampled through two steps
  ($60\times60 \rightarrow 120\times120 \rightarrow 240\times240$)
- $out_3$ (from D3): Upsampled once
  ($120\times120 \rightarrow 240\times240$)

The final output, `final_output`, is produced directly from the last decoder stage D4 at a resolution of $240\times240\times4$, serving as the primary segmentation map. Together, these four outputs provide a comprehensive basis for accurate tumor segmentation.

### 3.3.5 OUTPUT

The ARU-GD model produces four segmentation maps, the primary output, `final_output` (from D4), and three intermediate outputs, $out_1$, $out_2$, and $out_3$ (from D1, D2, and D3, respectively). Each output is a $240\times240\times4$ tensor that represents the probability distribution over the four tumor classes. This multi-map output facilitates both accurate segmentation and robust uncertainty estimation through Monte Carlo dropout.

## 3.4 DROPOUT INTEGRATION FOR UNCERTAINTY

Dropout layers with a rate of 0.2 are applied after the residual block in each decoder stage (D1 to D4). During training, dropout would randomly deactivate 20% of the neurons in each forward pass, introducing stochasticity to prevent overfitting and improve model generalization Srivastava et al. (2014) by reducing co-dependency among neurons. This regularization ensures that the model learns robust features across the decoder stages, which is critical for handling the complex and heterogeneous nature of brain tumor MRI data.

During inference, dropout is typically disabled in standard models to produce deterministic predictions. However, in the ARU-GD+MCD model, dropout is retained at inference time to enable uncertainty estimation via Monte Carlo dropout, a Bayesian approximation technique. This involves performing 20 forward passes with dropout enabled, generating a set of stochastic predictions for each pixel across the segmentation maps (`final_output`, $out_1$, $out_2$, $out_3$). Each forward pass yields a different segmentation map due to the random deactivation of neurons, simulating samples from the model's posterior distribution over the weights. The mean of these predictions provides the final segmentation map, while the variance or entropy of the predictions quantifies epistemic uncertainty capturing uncertainty in the model parameters themselves Gal & Ghahramani (2016). Uncertainty was quantified using predictive entropy across multiple passes Brosse et al. (2021). This approach is particularly valuable in medical imaging, as it highlights ambiguous regions (e.g., tumor boundaries) that may require further clinical evaluation, enhancing the reliability and interpretability of the segmentation results.

## 3.5 TRAINING DETAILS

The model was trained on the 4700 training slices using a batch size of 8 for 100 epochs, with 587 steps per epoch. Data augmentation (random rotations by $10°$, flips, shifts by 10%, shear by 10%) was used to make the model robust Shorten & Khoshgoftaar (2019). The 775 validation slices were checked with 96 steps per epoch. The model was trained on a system with an Intel i9 processor, 16 GB RAM, and an NVIDIA RTX 4070 GPU with 8 GB VRAM. The Adam optimizer (learning rate 0.0001) Kingma & Ba (2015) and a LossScaleOptimizer Micikevicius et al. (2018) were used. The loss combined weighted dice loss and log loss, defined as:

$$L_{\text{gen}} = L_{\text{weighted\_dice}} + L_{\text{weighted\_log}} \tag{2}$$

where the weighted dice loss component is given by:

$$L_{\text{weighted\_dice}} = 1 - \frac{\sum w_c \cdot \text{Dice}_c}{\sum w_c} \tag{3}$$

with weights $w_c = [1, 5, 2, 4]$, and $L_{\text{weighted\_log}}$ is a weighted categorical cross-entropy. The weighted Dice loss compensates for class imbalance Sudre et al. (2017). Loss weights were 0.5 for the main output and 0.125 for each extra output ($out_3$, $out_2$, $out_1$). The best model was saved based on training loss.

## 3.6 UNCERTAINTY MAP GENERATION.

Uncertainty maps were created using the dropout predictions. For each test sample, 20 forward passes with active dropout produced a set of probability maps. The mean probability, $p_{\text{mean}}(x)$, across these passes was calculated as:

$$p_{\text{mean}}(x) = \frac{1}{T} \sum_{t=1}^{T} p_t(x) \tag{4}$$

where $T = 20$ and $p_t(x)$ is the prediction for a given sample from pass $t$.

Predictive entropy, a measure of uncertainty, was computed as DeVries & Taylor (2018):

$$H(x) = -\sum_{c=0}^{3} p_{\text{mean}}(x, c) \log(p_{\text{mean}}(x, c)) \tag{5}$$

where the sum is over the four class channels ($c$). The resulting uncertainty maps range from 0 (low uncertainty) to 1 (high uncertainty). Displaying these maps alongside the segmentation output provides insight into where the model is more or less confident, supporting clinical interpretation and helping identify areas that may require further review Jungo & Reyes (2020)Nair et al. (2020).

## 4 RESULT

We evaluated our models on the BraTS dataset using 1,000 test slices. Performance was measured using Dice Score and Intersection over Union (IoU) across the three tumor regions: whole tumor (WT), tumor core (TC), and enhancing tumor (ET).

The detailed results are presented in Table 1 for Dice Scores and IoU Scores. As shown in Table 1, adding the Monte Carlo Dropout (MCD) layer to our model led to a significant improvement in the Dice scores for the TC and ET regions. This indicates that the model became much better at accurately segmenting these smaller and more complex tumor areas. Meanwhile, the performance on the WT region remained comparable to the best baseline model, demonstrating that the addition of MCD did not negatively affect the segmentation of larger tumor regions.

Table 1: Comparison of dice score and IoU for different models across tumor regions.

| Model | Dice score | | | IoU | | |
|---|---|---|---|---|---|---|
| | WT | TC | ET | WT | TC | ET |
| UNet | 0.858 | 0.822 | 0.700 | 0.752 | 0.699 | 0.539 |
| Res-UNet | 0.871 | 0.803 | 0.725 | 0.772 | 0.672 | 0.570 |
| AG Res-UNet | 0.893 | 0.857 | 0.761 | 0.807 | 0.750 | 0.614 |
| ARUNet+GD | 0.911 | 0.876 | 0.801 | 0.838 | 0.781 | 0.668 |
| **ARUNet+GD+MCD** | **0.886** | **0.899** | **0.856** | **0.793** | **0.818** | **0.748** |

Figure 2 shows sample input MRI slices alongside their ground truth, corresponding segmentation results and uncertainty maps predicted by our ARU+GD+MCD model. The model accurately delineates the tumor regions, highlighting its ability to capture both large and small tumor structures effectively.

Our model's final output gives not only the tumor segmentation but also uncertainty maps. These maps show which parts of the image the model is unsure about. This helps doctors see where the model's predictions might need a closer look. The uncertainty maps use colors from red to green to show how confident the model is. Red means high uncertainty, the model is not sure about these areas. Green means low uncertainty-the model is confident and likely correct there. This color coding makes it easy to spot parts of the tumor where the model might have made mistakes. Figure 2 shows examples of the uncertainty maps generated by our model. These were computed by aggregating stochastic predictions from multiple passes with dropout active, allowing the model to reflect epistemic uncertainty in regions of complex structure or sparse data Gal & Ghahramani (2016).

If you look closely at the uncertainty map in Figure 2, you'll see that it not only shows what the model predicted, but also how confident it is about those predictions. The green regions represent areas where the model is very sure, and these are also the areas where the model's predictions are correct. So, green means the model is confident and got it right.

Now, notice the red regions. These are where the model is not confident, and importantly, these are also the parts where the model has made mistakes, the predictions in the red areas are incorrect. These red zones mostly appear around the edges of the tumor, which makes sense because boundaries are often the trickiest to detect accurately. The tissue at the edges might be blurry or mixed with surrounding areas, making it harder for the model to decide what's what.

Such visualizations have been shown to be an effective way of communicating prediction reliability in clinical settings, helping clinicians prioritize which regions need the most attention during diagnosis and verification Jungo & Reyes (2020).

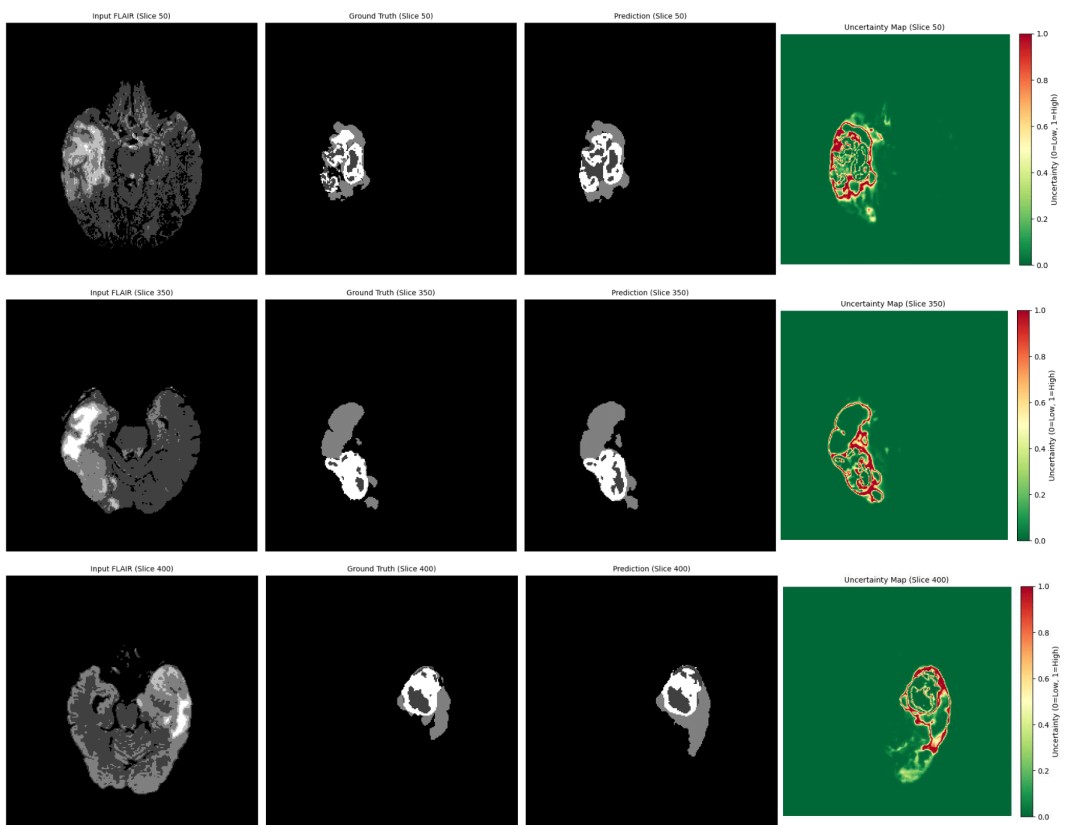

Figure 2: Sample images of Input MRI Scan, Ground Truth, Predicted and the Uncertainty Maps

## 5 DISCUSSION

Although newer versions of the BraTS dataset, such as BraTS 2021, 2022, and 2023 are available, we chose to train our model on BraTS 2019. Our objective was to build a model that provides uncertainty maps along with up-to-the-mark accuracy, rather than solely pushing for marginal gains on the latest benchmarks. Since the focus is on estimating uncertainty, especially around tumor boundaries, the specific dataset version does not significantly impact the validity of our approach. Moreover, BraTS 2019 remains a robust, well-annotated, and widely-adopted benchmark, making it well-suited for both segmentation and uncertainty evaluation tasks.

## 6 CONTRIBUTION

- Primary focus on uncertainty estimation:
  The main objective of our model is to generate uncertainty maps along with tumor segmentation. These maps highlight regions where the model is less confident, typically around tumor boundaries, helping doctors make better-informed decisions, improving treatment planning, and saving valuable time during manual verification.

- Improved accuracy with uncertainty integration:
  Even after integrating Monte Carlo dropout for uncertainty estimation, the model maintains strong segmentation performance. Accuracy is improved in tumor core (TC) and enhancing tumor (ET) regions, and remains comparable in the whole tumor (WT) region.

- Robust and consistent results:
  Trained on a large and well-annotated dataset, the BraTS 2019, the model demonstrates consistent and reliable performance across diverse MRI scans, enhancing its potential for real-world clinical use.

# 7 CONCLUSION

The main focus of this research was to introduce uncertainty estimation into brain tumor segmentation, allowing the model not only to predict tumor regions but also to indicate how confident it is in each part of its prediction. By generating uncertainty maps as a final output, our model helps identify areas of low confidence, especially around tumor boundaries which are often the most error-prone. This additional layer of information provides a practical advantage in medical settings, as it enables doctors to quickly focus on the uncertain regions, saving time and improving trust in automated segmentation.

To achieve this, we integrated Monte Carlo dropout into the architecture without sacrificing segmentation performance. In fact, our results showed that this addition not only preserved the model's accuracy but actually improved performance on challenging regions like the tumor core (TC) and enhancing tumor (ET), while maintaining strong results on the whole tumor (WT). Deep learning has brought significant advancements in medical image analysis Li et al. (2017).

Overall, the proposed ARG-UNet+MCD model offers both high-quality segmentation and reliable uncertainty estimation, making it a valuable and efficient tool for real-world clinical use.

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
