# OpenReview forum: "Uncertainty-Aware Brain Tumor Segmentation Using Attention Residual U-Net with Guided Decoder and Monte Carlo Dropout"
_ICLR.cc/2026/Conference — Submitted to ICLR 2026_

### Official Review · Reviewer_PhZe · 2025-11-01

**Soundness:** 3
**Presentation:** 2
**Contribution:** 2
**Rating:** 2
**Confidence:** 3

**Summary:**

The paper introduces a new model, ARU-GD+MCD, for the brain tumour segmentation task. It also employs this technique to estimate the uncertainty of segmentation predictions, thereby aiding in medical diagnosis. The evaluation shows that ARU-GD+MCD marginally outperforms the discussed benchmarks.

**Strengths:**

- The plan to use ARU-GD with Monte Carlo Dropout is novel for estimating the uncertainty for segmentation.

- There are enough details about the methodology in the paper, facilitating reproducibility.

**Weaknesses:**

- Very limited discussion of prior work/techniques used for brain tumour segmentation uncertainty, other than in the context of Bayesian uncertainty. There aren't enough motivations and references to previous works in the introduction to establish the shortcomings of the current literature and motivate the need for this technique.

- Most of the details of the Unet covered in the paper are quite basic and do not provide any novelty under section 3.3. These details, if required, would be better accommodated under a section in the Appendix.

- Presentation: (1) Please follow a consistent citation style, (2) Maintain consistent notations. For example, ARUNet+GD+MCD and ARUNet-GD+MCD have been used interchangeably, and (3) it would be helpful to give citations in the table for other benchmarks.

- There is not much novelty. Both the ARU-GD backbone and Monte Carlo dropout are established; the contribution mainly lies in combining them effectively.

- The results are presented for a single dataset. As the paper promises a new technique for the brain tumour segmentation task, evaluation against multiple datasets would add to the paper's value, making it more useful to a larger community.

**Questions:**

- Can the authors briefly describe how Monte Carlo Dropout will account for showing epistemic uncertainty and not stochastic/aleatoric? Providing the same input to the model multiple times could lead to different outputs due to the use of different seeds for adding noise, etc., in the Unet, which can account for aleatoric uncertainty.

- In line 253, you mention a class imbalance among the 4 modalities. However, the authors mention that the input to the Unet is always all 4 modalities for every input. Could you please elaborate more on the scaling aspect while computing the dice loss?

---

### Official Review · Reviewer_HEuk · 2025-11-02

**Soundness:** 1
**Presentation:** 2
**Contribution:** 1
**Rating:** 0
**Confidence:** 4

**Summary:**

The paper combines a known architecture (Attention-ResUNet with guided decoder) and uncertainty estimation technique (specifically MC Dropout) in deep learning to show improved performance on brain tumor segementation task (BraTS 2019 Dataset) compared to vanilla UNet family (UNet, ResUNet, Attention UNet). Visual results show outputs from such uncertainty estimation technique allow not only class labels but also lesion regions that the model is not sure about. The paper advocates that this can be a useful tool for clinical application.

**Strengths:**

The paper is fairly easy to read and details for reproducibility are well detailed. The paper focuses on the challenge of brain tumor segmentation, especially with the large 3D volume inputs (4 separate MRI volumes per patient) and the difficulty of the task (segmentation of 3 separate tumor subtypes) which is commendable, well motivated and significant.

**Weaknesses:**

*Weak Contribution*: The proposal seems to combine already proposed ideas in a trivial manner, resulting in a very weak contribution. Both ARUnet+GD and MCDropout have been proposed elsewhere, and it is known that adding uncertainty estimation and ensembling methods improves model performance across a wide range of architectures.

*Data Preprocessing*:Using only 25 mid-slices out of 155 is unjustified. This prioritizes computational efficiency over clinical validity, as lesions outside these slices are ignored - potentially missing clinically important regions. If applied to both training and validation, this limits clinical relevance; if only to training, it introduces domain shift. Both cases undermine the study’s validity.

*Results*: No confidence intervals or statistical significance tests are reported, making it unclear whether the performance improvements are meaningful.

**Questions:**

*Contribution*: The contributions combines previously established methods (MC Dropout and Guided decoder losses) in a routine way, offering little novelty. The paper, in the state as it is, could be a good starting point to explore rigorous evaiuation of architectures and uncertainty estimation.

*Data Preprocessing*: The MRI Volumes have been oddly preprocessed with only 25 (mid)slices out of 155 slices taken as input with the rationale given “to focus on key tumor cases and reduce processing needs”. This rationale is focused on reducing processing at the cost of a somewhat clinically catastrophic decision. In Brain tumor segmentation, it may be important to be able to segment/detect each individual lesion not just improve perfomance at global image level. BraTS 2023 has even introduced a new metric – Lesionwise DSC – that focuses on exactly this. Only training on subset of core slices and ignoring lesions in regions outside of these core slices seem to misplace attention on computational efficiency at the cost of clinical relevance.

It is also unclear to me if the same preprocessing is done for validation as well. If yes, then it ignores lesions outside of this 25-slice volume. If not, then this results in a domain shift between training and validation set where the validation set gets to see the whole brain while the training set does not. Hence, this seems problematic in both cases.

*Writing*: Since the ARU-GD architecture has been proposed elsewhere, one may point to the original paper for details. The almost 2 page detailed explanation of the architecture seem redundant, the space which could have been used for extensive analysis instead.
In general, the writing could have been made better with omission of superfluous adjectives (e.g. ARU-GD is an effective deep learning model) and made less verbose.

The citation formatting seem incorrect. Additionally, The citation of the ARU-GD paper seem incorrect. [Maji 2020 -> Maji 2022]
in the paper: Debesh Maji, Chandrima Chakraborty, and Sayan Paul. Attention Res-UNet with guided decoder for semantic segmentation of brain tumors, 2020. Conference paper, details not specified.

Probably should have been: Maji, Dhiraj, Prarthana Sigedar, and Munendra Singh. "Attention Res-UNet with Guided Decoder for semantic segmentation of brain tumors." Biomedical Signal Processing and Control 71 (2022): 103077.

---

### Official Review · Reviewer_6LZV · 2025-11-07

**Soundness:** 1
**Presentation:** 2
**Contribution:** 1
**Rating:** 2
**Confidence:** 5

**Summary:**

This manuscript presents an investigation into uncertainty-aware brain tumor segmentation by augmenting a known architecture, the Attention Res-UNet with Guided Decoder (ARU-GD), with Monte Carlo (MC) dropout layers. The resultant model, ARU-GD+MCD, aims to address the critical gap between high segmentation accuracy and clinical utility by providing uncertainty maps to guide clinician review. By retaining dropout during inference, the model approximates Bayesian inference, allowing for the quantification of epistemic uncertainty via predictive entropy. The primary contribution is posited as the generation of this "interpretability layer" to enhance diagnostic efficiency and improve trust in automated segmentation tools.

**Strengths:**

1. The script is founded on a robust and clinically relevant problem statement.

**Weaknesses:**

1. Lack of methodological novelty. The core technique, applying MC dropout to a U-Net-based architecture for uncertainty in medical imaging, is not novel. This approach has been widely adopted. The base ARU-GD architecture is also acknowledged as pre-existing work.
2. The benchmarking is insufficient. One of the state-of-the-art frameworks for biomedical image segmentation, nnU-Net [1], was not compared.
3. The literature review (Section 2.2) explicitly mentions alternative, and in some cases more sophisticated, approaches to uncertainty quantification, such as variational inference, auto-encoding variational Bayes, and variational dropout with local reparameterization. However, the study implements and evaluates only the MC dropout
4. The paper’s central claim is the clinical utility of its uncertainty maps. However, the validation of these maps is entirely qualitative, relying on a visual assessment of Figure 2. A robust validation of uncertainty requires quantitative metrics. The paper fails to report calibration curves (reliability diagrams), Expected Calibration Error (ECE), or any statistical analysis correlating predictive entropy with segmentation accuracy (e.g., Dice score vs. uncertainty quantile).
5. The reliance on the BraTS 2019 dataset for a 2026 conference submission is a notable limitation. Datasets like BraTS 2021-2023 are available.
6. The proposed method requires $T=20$ stochastic forward passes to generate one prediction and its corresponding uncertainty map.1 This introduces a $\approx$20-fold increase in inference time compared to a deterministic model. The manuscript fails to discuss, or even mention, this substantial computational cost.

[1] Isensee, Fabian, et al. "nnU-Net: a self-configuring method for deep learning-based biomedical image segmentation." Nature methods 18.2 (2021): 203-211.

**Questions:**

1. Does the paper provide any quantitative validation of this correlation (e.g., Expected Calibration Error, reliability diagrams, or an analysis of Dice scores versus predictive entropy) to scientifically substantiate the reliability of its uncertainty estimates?
2. How does the method address the substantial computational overhead?

---

### Meta-Review · Area_Chair_VuZC · 2025-12-04

**Summary:**

All reviewers tend to reject the paper, and the author did not provide any response, so I tend to reject the paper.

**Reviewer Concerns:**

All three reviewers believe that the paper merely concatenates the "existing ARU-GD architecture" with the "existing MC-Dropout technique" in a simplistic manner, lacking novel algorithmic or theoretical contributions; no mechanistic explanation is provided for why this combination can enhance segmentation accuracy.
Furthermore, all reviewers emphasized that the paper only presented a visual uncertainty map, without reporting a calibration curve or conducting statistical correlation analysis between "high uncertainty areas" and "segmentation error areas". Therefore, it is impossible to scientifically prove that the uncertainty estimation is truly reliable.
Many people have pointed out that the results lack persuasiveness due to the lack of comparison with the strong baseline of current medical image segmentation, nnU-Net, and the absence of experiments on updated datasets such as BraTS 2021-2023.
The author did not provide any response during the rebuttal period.

**Reviewer Scores:**

The author did not provide any response. Therefore, I believe the reviewers will not change their scores.

---

### Decision · Program_Chairs · 2026-01-26

Reject